# Position: Collaborative Agentic AI Needs Interoperability Across Ecosystems

Rishi Sharma [1] [2]    Martijn de Vos [1]    Pradyumna Chari [2]    Ramesh Raskar [2]    Anne-Marie Kermarrec [1]

## Abstract

Collaborative agentic AI is projected to transform entire industries by enabling AI-powered agents to autonomously perceive, plan, and act within digital environments. Yet, current solutions in this field are all built in isolation, and we are rapidly heading toward a landscape of fragmented, incompatible ecosystems. In this position paper, we argue that **interoperability, achieved by the adoption of minimal standards, is essential to ensure open, secure, web-scale, and widely-adopted agentic ecosystems**. To this end, we devise a minimal architectural foundation for collaborative agentic AI, named WEB OF AGENTS, which is composed of four building blocks: agent-to-agent messaging, interaction interoperability, state management, and agent discovery. WEB OF AGENTS adopts existing standards and reuses existing infrastructure where possible. With WEB OF AGENTS, we take a first but critical step toward interoperable agentic systems and offer a pragmatic path forward before ecosystem fragmentation becomes the norm.

## 1. Introduction

Agentic artificial intelligence (AI) is an emerging approach in which autonomous agents, typically powered by large language models (LLMs), can make decisions, take actions, and learn from the environment in which they are deployed (Acharya et al., 2025). The capabilities of such agents go beyond generative AI as they can invoke tools, maintain long- and short-term memory, and engage in complex workflows with other agents (Liu et al., 2025; Murugesan, 2025). Their ability to autonomously plan, communicate, and collaborate unlocks a new class of possibilities ranging from scientific discovery to multi-agent enterprise orchestration (Ghareeb et al., 2025; Li et al., 2023; Lu et al.,

2024; Park et al., 2023). As academia and industry starts to understand these systems better, the impact of agentic AI is poised to be transformative.

While individual agents have shown tremendous potential with the use of tools such as deep research (OpenAI, 2025) and web search (Perplexity, 2025), multi-agent AI systems have recently gained traction to enable *collaborative agentic AI* (Liu et al., 2025; Sapkota et al., 2025). Recent deployments demonstrate their potential across scientific domains: AI Scientist (Lu et al., 2024) coordinates specialized agents for end-to-end research automation, ROBIN (Ghareeb et al., 2025) integrates planning and experimental agents for biomedical discovery, and comparable systems accelerate materials discovery through coordinated simulation and modeling agents (Bazgir et al., 2025). As these agentic capabilities proliferate across domains, the next frontier is a decentralized WEB OF AGENTS where agents can dynamically discover each other's expertise and establish communication to tackle complex, cross-disciplinary challenges that no single agent could address alone (Malka et al., 2024; Shlezinger et al., 2021).

Currently, there is a significant effort to build infrastructure for collaborative agentic AI. Notable examples include the Agent2Agent (A2A) protocol (Google DeepMind, 2025), released by Google in April 2025, the Model context protocol (MCP) (Model Context Protocol Project, 2025), released by Anthropic in November 2024, and the Agent communication protocol (ACP) (Agent Communication Protocol Project, 2025), released by IBM in April 2025 (merged into A2A later). These solutions introduce necessary primitives such as agent-to-agent communication protocols, agent authentication methods, and mechanisms for expressing the capabilities of agents. However, these siloed efforts develop in isolation with distinct abstractions and data formats, creating a fragmented landscape where agents from one ecosystem cannot interoperate or interact with agents from another. We visualize this challenge in Figure 1 (left). While such diverse and decentralized efforts in implementation accelerates innovation and experimentation (Eiras et al., 2024; Raymond, 1999), our concern is that the resulting ecosystem fragmentation imposes engineering overheads, limits reusability, and threatens long-term scalability and security.

Two obvious approaches could address this fragmentation:

[1]EPFL, Lausanne, Switzerland [2]Massachusetts Institute of Technology, Cambridge, MA, USA. Correspondence to: Rishi Sharma <rishi.sharma@epfl.ch>.

*Proceedings of the 43rd International Conference on Machine Learning*, Seoul, South Korea. PMLR 306, 2026. Copyright 2026 by the author(s).

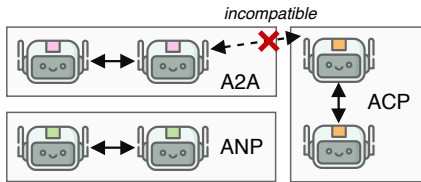 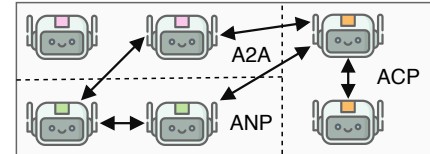

*Figure 1. Left:* Prominent protocols for collaborative agentic AI (A2A (Google DeepMind, 2025), ANP (Agent Network Protocol Project, 2025) and ACP (Agent Communication Protocol Project, 2025)). Agents by default cannot communicate with agents in other ecosystems, resulting in fragmentation (see Section 3). *Right:* our proposed solution, WEB OF AGENTS.

*(i)* enforcing a single unified protocol, or *(ii)* creating post-hoc translation layers between each pair of systems. However, history suggests that both approaches are problematic. The former stifles innovation and often fails to gain adoption, as seen with the OSI model's struggle against TCP/IP's pragmatic success (Clark, 1988; Leiner et al., 1997; 2009; Russell, 2014), while the latter creates undesirable complexity and maintenance burdens (Berners-Lee, 1999; Bollinger, 2000; Chen et al., 2023; Kolos, 2000; Zhao et al., 2023). In this position paper, we argue that the field of collaborative agentic AI needs a third approach to address fragmentation. We strongly advocate for a WEB OF AGENTS: an interoperable ecosystem of collaborative agents that are grounded in minimal standards. We visualize this vision in Figure 1 (right). More concretely, **our position is that interoperability, achieved by the adoption of minimal standards, is essential to ensure open, secure, web-scale, and widely-adopted agentic ecosystems**. Rather than constraining development by enforcing a particular protocol or creating brittle translation layers, we advocate for the adoption of lightweight standards that already power web infrastructure. This interoperability enables several key benefits: reduced duplication of effort, scalable ecosystem growth, enhanced security through shared scrutiny, and greater autonomy for users and agent operators as they can freely switch between agents that provide similar services.

To realize this vision, we identify four minimal, yet sufficient building blocks that ecosystems should adopt for enabling interoperability: *(i)* agent-to-agent messaging, enabling asynchronous and structured communication via existing web protocols, *(ii)* interaction interoperability, establishing shared interaction specifications through interaction documents, *(iii)* state management, supporting both short-term sessions and long-term state, and *(iv)* discovery, allowing agents to find one another. Our approach requires minimal new infrastructure, just careful extensions of the web as we know it. Together, these building blocks form the foundation of the WEB OF AGENTS.

**Contributions.** To defend our position, we first analyze several prominent solutions in Section 3, identifying common trends across current frameworks for collaborative agentic AI. This analysis motivates our arguments in Section 4 that

interoperability is not merely desirable, but essential for realizing the full potential of collaborative agentic AI. We then propose in Section 5 a minimal blueprint, WEB OF AGENTS, that demonstrates how this interoperability can be achieved through careful reuse and extension of existing web technologies. We open-source our implementation of applying the standards to A2A, ACP, and Agora, including a proof-of-concept implementation (see Section 5.6).Finally, we discuss some views that oppose our position in Section 6 before concluding with final remarks in Section 7.

We note that some of the web technologies we advocate for are decades old and deeply embedded in the Internet infrastructure. This is precisely the point we want to make. Our contribution is not the invention of new protocols, but the recognition that the AI community does not need to reinvent them to realise collaborative agentic AI. Much as the internet architecture community learned through hard-won experience, we argue that minimal, battle-tested standards are the right foundation. This is a learning that has yet to be absorbed by the rapidly evolving field of collaborative agentic AI.

## 2. From Generative AI to Collaborative Agentic AI

**Generative AI.** Generative AI encompasses computational models capable of producing novel content across multiple modalities, including text, images, and audio (Feuerriegel et al., 2024). Among the most prominent implementations is ChatGPT, which provides a conversational interface to large language models (LLMs) through natural language processing (Achiam et al., 2023). LLMs demonstrate proficiency in diverse language generation tasks, including document summarization, content synthesis, and creative text generation. The standard generative AI workflow, illustrated in Figure 2 (left), follows a request-response pattern where users submit queries to an LLM system (Step 1) and receive generated outputs (Step 2). This architecture supports multi-turn conversations, enabling contextual dialogue through sequential query-response exchanges within a persistent session.

However, this conventional user-to-LLM paradigm exhibits notable architectural limitations (Schneider, 2025). A fun-

damental constraint is the models' inability to perform autonomous environmental observation, external system interaction, or action execution without explicit human orchestration. Such capabilities would encompass database querying, web navigation (Drouin et al., 2024), or programmatic transaction execution. While retrieval-augmented generation (RAG) frameworks (Lewis et al., 2020) partially address these limitations by augmenting model inputs with relevant external information, these approaches typically require separate orchestration mechanisms to manage the retrieval process.

**Agentic AI.** To address these architectural constraints, the field has witnessed a paradigm shift toward agentic artificial intelligence systems (Hughes et al., 2025; Murugesan, 2025). Agentic AI architectures are fundamentally structured around autonomous *agents*—computational entities capable of independent operation and dynamic interaction with external *tools*, as illustrated in Figure 2 (middle). These agents are typically specialized and optimized for specific task domains, such as computer vision processing or web navigation protocols. Tools constitute external service interfaces that agents can programmatically invoke to execute functions beyond the inherent capabilities of the underlying LLMs. Tool selection and orchestration represent critical components of the response generation pipeline, requiring systematic evaluation of available resources for optimal task completion (Ruan et al., 2023). Agents can autonomously invoke multiple tools in parallel (Step 2 in Figure 2) when required, utilizing tool responses to inform subsequent planning and action selection. These tools maintain direct interfaces to various data repositories and external systems (Step 3). Agentic AI frameworks demonstrate significant potential for transformative applications across domains including healthcare and software engineering (Murugesan, 2025).

However, contemporary agentic architectures predominantly operate through single-agent paradigms, constraining system scalability, modularity, and fault tolerance. This architectural limitation necessitates that individual agents manage comprehensive task execution, logical reasoning, and tool orchestration, potentially exceeding agent capabilities when confronting complex scenarios requiring specialized domain expertise or access to heterogeneous tool ecosystems.

**Collaborative Agentic AI.** Collaborative agentic AI represents the next step and involves AI agents that can communicate and collaborate across tasks, domains, and organizational boundaries. We visualize this frontier in Figure 2 (right). Compared to agentic AI with a single agent, agents can form collectives and work together (Step 2). For example, a travel agent can coordinate with a calendar agent to check availability, a payment agent to process transactions, and a policy agent to enforce organizational constraints.

These interactions can all occur without human intervention. Such networks of LLM-based agents have already been used to simulate human and economic behavior (Li et al., 2023; Park et al., 2023), and to understand dynamics in societies (Gürcan, 2024).

The recent boom in AI agents follows substantial work on multi-agent systems (MAS), a field with decades of research on autonomous agents that, similar to collaborative agentic AI, coordinate, negotiate, and act in digital environments (Ferber, 1999; Finin et al., 1994; FIPA, 1997; 2002). Classical MAS research focuses on pre-programmed agents with static rules, whereas LLM-powered agents can operate in more unstructured environments and reason using natural language. Therefore, while we acknowledge the field of classical MAS, we position this work towards LLM-powered agents. Beyond classical MAS, the multi-agent reinforcement learning (MARL) literature has also contributed foundational ideas to agent coordination, communication, and emergent collaboration (Zhang et al., 2021) However, our work operates at a different level: while MARL addresses how agents learn to coordinate, we focus on the systems-level challenges of cross-ecosystem collaboration among diverse LLM agents at web scale.

## 3. The Alarming Trend Toward Incompatible Agentic Ecosystems

The potential of (collaborative) agentic AI has resulted in significant research and engineering efforts from both industry and academia to devise new infrastructure, protocols, and frameworks (Hosseini & Seilani, 2025; Tran et al., 2025). Notable examples include Google's A2A protocol (Google DeepMind, 2025), Anthropic's MCP (Model Context Protocol Project, 2025), and IBM's ACP (Agent Communication Protocol Project, 2025), leading to the "protocol wars" phase of (collaborative) agentic AI. Based on existing work (Ehtesham et al., 2025) and our own research, we identify prominent solutions for (collaborative) agentic AI and look at their innovation areas, see Table 1. Even though most solutions focus on inter-agent communication, we include MCP and agents.json in our analysis because of their increased adoption in agentic workflows (Ehtesham et al., 2025). The goal of our analysis is not to point out particular shortcomings of individual solutions but rather to take a broader perspective on the development of agentic AI solutions.

Our main observation is that **agentic AI solutions are developed in isolation and are incompatible with each other.** As a result, we are heading toward a collection of ecosystems in which agents built using one framework are unlikely to interoperate with those from another (Research, 2024). Such fragmentation introduces inefficiencies, for example, companies wishing to deploy agents that interact across systems must either commit to a single protocol or reimplement

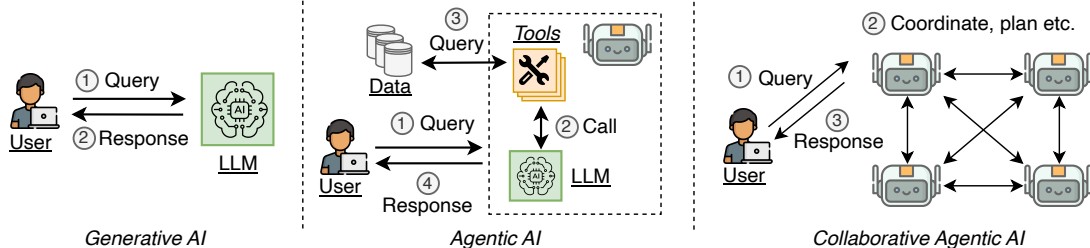

*Figure 2.* Generative AI (left), agentic AI (middle), and collaborative agentic AI (right). This work provides a blueprint for interoperable collaborative agentic AI that leverages existing web protocols.

agents for each distinct ecosystem. As new solutions continue to emerge, these interoperability challenges are poised to grow.

One might address the interoperability challenges among diverse agentic AI systems with *translation layers* or *mediators* that convert messages and actions between different protocols (Paniagua, 2021). This approach, for example, has been adopted to connect fragmented web standards in the early days of the web and to enable value exchange between different blockchain platforms. While this brings interoperability, the viability of translational layers has been questioned (Berners-Lee, 1999; Kolos, 2000). Firstly, translation layers can become brittle or outdated as protocols evolve. Secondly, such layers can expand the system's attack surface and expose vulnerabilities that malicious actors could exploit, as has been observed with cross-blockchain bridges (Zhao et al., 2023). Thirdly, as the number of distinct protocols increases, maintaining translation mappings becomes increasingly difficult (Chen et al., 2023). These challenges raise serious concerns about whether translation layers can serve as a long-term solution for agentic AI interoperability.

Beyond interoperability challenges, this ecosystem-level isolation has also led to uneven progress across the agentic AI stack. A holistic solution for agentic AI requires, among others, primitives such as agent identity, protocols for inter-agent communication, agent discovery mechanisms, and mechanisms for memory management. We observe in Table 1 discrepancies in the focus of innovation areas across solutions. On the one hand, all identified solutions emphasize some notion of agent identity, such as proposing mechanisms for agents to describe their capabilities or roles. On the other hand, only a minority of solutions implement mechanisms for the discovery of other agents in the network. Similarly, the capability of an agent to manage short- or long-term memory receives inconsistent attention across solutions. This uneven focus results in fragmented tooling that hinders composability across systems. This pattern is not unique to agentic AI: similar fragmentation has occurred in domains such as internet-of-things (Aly et al., 2018), blockchain (Zamyatin et al., 2021), and early web

services (Papazoglou & Van Den Heuvel, 2007), where numerous tools emerged to solve narrow problems in isolation.

Some analyzed solutions introduce their own implementations of core primitives such as authentication, capability exposure, agent discovery, and message exchange formats. For example, while some leverage existing authentication schemes such as the OAuth protocol (Hardt, 2012) or decentralized identifiers (DIDs) (Sporny et al., 2022), others either rely on custom mechanisms or leave this aspect unspecified. Similarly, agent discovery is often realized through custom listings with their own formats. Generally, the result is a patchwork of ecosystems that are incompatible by design. Moreover, some protocols introduce new primitives without clearly defined use cases, optimizing for hypothetical scenarios rather than concrete needs. LMOS, for example, follows a maximalist design by building a complete OS-like environment for LLMs. Messaging applications are a pertinent analogy to this observation: major services like iMessage, WhatsApp, Signal, or Telegram all implement their own protocol stack, message format, and identity systems, and are largely incompatible with each other (Arnold & Schneider, 2017).

The field of agentic AI stands at a critical point. In the absence of coordination, it risks splintering into competing silos, each solving similar problems in slightly different and incompatible ways. For instance, consider a travel-booking agent developed by a major airline that supports rich capabilities such as real-time seat selection using a particular agent protocol. Even if this agent is exceptionally capable, users and other agents relying on a different agentic ecosystem would be unable to invoke it or compose it into larger workflows. In this way, incompatibility directly burdens engineers and diminishes the quality of service for users.

> **Key Takeaway**
>
> Because each solution is being developed in isolation, agentic AI is evolving into a landscape of incompatible ecosystems.

*Table 1.* Prominent solutions for (collaborative) agentic AI and their innovation areas.

| Solution | Agent Identity | Inter-Agent Comm. | Agent Discovery | State Management | Interoperability w/ other solutions |
|---|---|---|---|---|---|
| **A2A** (Google DeepMind, 2025) | ✓AgentCard | ✓Async. | ❖Static | ✗ | ✗ |
| **MCP** (Model Context Protocol Project, 2025) | ❖ Contextual | ✗ | ✗ | ✗ | ✗ |
| **ACP** (Agent Communication Protocol Project, 2025) | ✓ | ✓REST API | ✗ | ✗ | ✗ |
| **agents.json** (Wildcard AI, 2025) | ❖ Capabilities | ✗ | ✗ | ✗ | ✗ |
| **Agora** (Marro et al., 2024) | ❖Protocol docs | ✓ | ✗ | ✗ | ✗ |
| **ANP** (Agent Network Protocol Project, 2025) | ✓Identity Layer | ✓ | ✓ | ✗ | ✗ |
| **AITP** (NEAR AI, 2025) | ✓ | ✓Chat Threads | ✗ | ✓ | ✗ |
| **LMOS** (Eclipse Foundation, 2025) | ✓ | ✓ | ✓ | ✓ | ✗ |

# 4. Interoperability as Necessary Foundation for Agentic AI

If fragmentation is the current trajectory, interoperability is the antidote. To achieve this, we advocate for the adoption of **minimal standards** that enable agents to interact across different ecosystems. Minimal standardization has historically enabled breakthrough innovations across domains: protocols like HTTP and HTML bootstrapped the web (Berners-Lee, 1999), SMTP and IMAP enabled global email interoperability (Crispin, 2003; Klensin, 2008), and TCP/IP continues to serve as the backbone of the Internet (Postel, 1981). With minimal, interoperable foundations, we strongly believe we can bootstrap a successful agentic ecosystem. Next, we further motivate interoperability and elaborate on how it strengthens agentic AI in terms of security, open participation, and scalability.

**Security.** Interoperability through minimal standardization contributes to the security of the overall ecosystem (Elkhodr et al., 2016). This standardization creates opportunities for systematic auditing, formal verification, and community-wide threat modeling. In fact, security is a critical concern in the emerging landscape of agentic AI and is a necessity to protect against adversarial agents or users (Shavit et al., 2023). This is because, unlike traditional systems that follow predefined logic, AI agents operate in dynamic environments, interpret ambiguous instructions, and make autonomous decisions, often without human supervision. This autonomy introduces novel attack surfaces, *e.g.*, prompt injection and jailbreak attacks (Liu et al., 2024; Wei et al., 2023). As these agents become embedded in critical workflows, security becomes paramount.

**Open Participation.** Interoperability empowers agent builders, *e.g.*, developers, organizations, or infrastructure providers, to exercise meaningful choice in the agents they interact with (Kapoor et al., 2025). In a collaborative agentic AI system, agents often depend on other agents. With shared protocols and minimal standards in place, builders can freely choose among competing agents that offer similar functionality but differ in trust models, performance, cost,

or affiliation. Without such freedom of choice, switching costs rise dramatically as each agent may require custom APIs, data formats, and authentication schemes. This not only leads to vendor lock-in, but also stifles innovation by discouraging new entrants from building compatible alternatives (Yan & Feng, 2023). This also holds for end users, who may want to directly use the services of agents across ecosystems or combine agents from different providers into a single workflow. An interoperable agentic AI ecosystem fosters open participation from both engineers and end-users, enabling vibrant competition, lowering barriers to entry, and promoting long-term ecosystem health (Aly et al., 2018).

Closely related, interoperability is also a prerequisite for a decentralized agentic AI ecosystem. Without a shared set of standards, coordination inevitably gravitates toward a small number of centralized platforms that dictate how agents are built and deployed. In contrast, minimal interoperability standards allow diverse actors to independently develop and operate agents, while still participating in the overall ecosystem (Kapoor et al., 2025). With interoperability, anyone can publish and run agents that are discoverable and usable by others without needing permission from any authority.

**Scalability.** Interoperability is also a key enabler of scalability in terms of ecosystem expansion. As the agentic AI landscape grows, no single provider will be able to build and maintain every agent, tool, or integration needed to support diverse user needs across domains. A scalable ecosystem depends on the ability of independent developers to contribute agents, services, and infrastructure components that can plug into a larger system without needing to coordinate closely with every other participant. Shared protocols reduce the integration overhead, allowing developers to focus on functionality rather than glue code. This mirrors the success of the web, where interoperable standards enabled a vast ecosystem of independent websites, browsers, and services to emerge without centralized planning (Berners-Lee, 1999). Furthermore, it allows the protocols to grow with the necessary improvements as the field evolves and particular use-cases become more prominent.

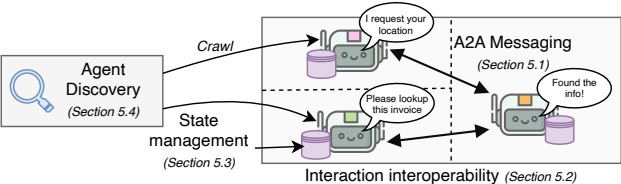

*Figure 3.* Blueprint of WEB OF AGENTS and its building blocks.

> **Key Takeaway**
>
> Interoperability is not just a nice-to-have feature, it is foundational. It enables secure, composable, scalable, and decentralized agentic ecosystems across domains.

## 5. Architectural Foundations for an Interoperable Web of Agents

We now identify a minimal set of building blocks and devise a blueprint for an interoperable agentic AI ecosystem named WEB OF AGENTS. The WEB OF AGENTS architecture is visualized in Figure 3, and the building blocks are summarized in Table 2. This blueprint adheres to the long-standing system design principle: *Keep It Simple, Stupid* (Lampson, 1983). This principle guides our strategy for enforcing *minimal standardization*, and *adopting existing standards* where possible. We strongly believe that the greatest interoperability can be achieved by staying as close as possible to existing web technologies. To reinforce our arguments, we discuss the proof-of-concept implementation for WEB OF AGENTS in Section 5.6. We remark that the building blocks we propose are application-level interoperability functions, not new network layers: they reuse existing web infrastructure rather than extending the networking stack.

### 5.1. Agent-to-Agent Messaging

Standardized message exchange protocols are essential for interoperability in an agentic AI ecosystem, as they establish the communication channels for coordination, negotiation, and collaboration. Without common messaging standards, each pair of interacting agents would require custom integration, creating an scaling problem that would severely restrict ecosystem growth (Papazoglou & Van Den Heuvel, 2007).

**Leveraging Hypertext Transfer Protocol (HTTP) for agent-to-agent communication.** HTTP, the protocol that underpins the modern web, is well-suited for agent-to-agent communication through its request-response pattern and well-defined semantics (Fielding & Reschke, 2014). By adopting HTTP as the foundational transport protocol, agents can exchange information using established meth-

ods: GET requests to retrieve information from other agents (similar to browsers accessing webpages), and POST requests for more complex interactions with messages and other context data in the payload. This HTTP-based approach leverages decades of rethinking, improvements, and optimizations without duplication of efforts (Mogul, 2002). While some emerging protocols like ACP and Agora directly leverage HTTP requests for agent-to-agent communication, other solutions like A2A have opted for more specialized protocols like JSON-RPC (Google DeepMind, 2025; Marro et al., 2024). We strongly believe that agent-to-agent communication does not require specialized protocols like JSON-RPC or WebSockets. While these protocols offer some compelling features such as asynchronous communication, enforcing these as a standard would be an overkill, hinder progress, and limit interoperability (Murley et al., 2021).

Another advantage of utilizing HTTP requests for agentic communication is that agents can coexist with webpages, making it trivial to integrate agentic AI systems with existing software. Furthermore, specialized patterns like asynchronous requests can be easily implemented on the client-side using HTTP requests and have long existed on the web (Microsoft, 2025). Finally, by building upon HTTP's extension mechanisms, this foundation allows for future enhancements without breaking backward compatibility, creating a stable yet evolvable communication layer for the WEB OF AGENTS.

Finally, adopting HTTP as the agent-to-agent transport layer inherits a mature set of security and privacy mechanisms. For example, TLS (Rescorla, 2018) provides end-to-end encryption of agent communications and OAuth 2.0 (Hardt, 2012) enables fine-grained authentication and authorization between agents. These are battle-tested mechanisms that receive far more community scrutiny than the custom security schemes introduced by individual agentic frameworks. Collaborative agentic AI also introduces privacy challenges beyond those of traditional web services, such as sensitive user data flowing between agents without explicit consent (He et al., 2025). This is an important open problem and we advocate for building privacy solutions for collaborative agentic AI using HTTP's existing foundations where possible.

### 5.2. Interaction Interoperability

Communication between autonomous agents presents unique challenges beyond basic message transportation. Agents are currently being developed using diverse frameworks, architectural approaches, underlying technologies, and with varying capabilities (see Section 3). In fact, different agents might need completely different inputs, possibly in different formats, to carry out their tasks. For example, a

*Table 2.* WEB OF AGENTS building blocks.

| Building block | Functional needs | Web technologies |
|---|---|---|
| Agent-to-agent messaging (Section 5.1) | HTTP-based messaging | HTTP requests |
| Interaction interoperability (Section 5.2) | Interaction documentation | API documentation |
| State management (Section 5.3) | Short-term memory | Sessions |
| | Long-term memory | DB integration |
| Agent discovery (Section 5.4) | Unique endpoints | URLs, DNS |
| | Capability advertisement | Well-known paths |

weather forecasting agent may expect location coordinates and a timestamp, while a financial analysis agent requires historical market data and risk parameters. Existing solutions typically do not sufficiently consider differences in agent design and capabilities. Protocols typically assume a universal interaction pattern without accounting for variability in inputs, expected conditions, or optimal message compositions that can be efficiently interpreted by the underlying LLM (Agent Communication Protocol Project, 2025; Google DeepMind, 2025). Therefore, collaborative agentic AI requires a way to enable interoperability in agent interactions.

**Interaction document.** To enable interaction interoperability, we propose an *interaction document* exchange between agents. This way, each agent can explicitly document its interface requirements, capabilities, expected inputs, outputs, and conditions necessary for effective communication. Crucially, this documentation need not follow a schema or predefined structure, where we differ from the classical MAS works such as KQML and FIPA (Finin et al., 1994; FIPA, 1997). Since modern AI agents are powered by LLMs, they can interpret and adapt to various documentation formats, from structured schemas such as JSON to natural language descriptions. This documentation acts as a standardized handshake, accessible via common endpoints using HTTP requests, and enables agents to understand and adapt to diverse interaction patterns without prior agreements or custom engineering efforts by the agent operator. This approach is minimalistic in the sense that we neither restrict how agents interact with each other nor impose rigid formatting requirements on their capability descriptions. Unlike traditional software systems that typically require specific schemas and parsers, agents can interpret unstructured interaction documents, written in natural language. We therefore envision a spectrum of interaction document formats: agents may use structured formats when precision is critical, while simpler interactions may rely on natural language descriptions, similar to how humans communicate. If ambiguity in a natural language document causes a failed interaction, agents can ask clarification questions to the counterparty agent in natural language.

This minimalistic approach to interaction interoperability offers three advantages. First, it enables agents to evolve independently, *i.e.*, the internal implementations can change completely as long as the documentation interface remains stable. Second, it fosters innovation by avoiding premature and overly restrictive standards, providing agent operators flexibility in their implementations. Ultimately, this straightforward yet effective solution mitigates the friction associated with agent integration.

### 5.3. State Management

Agent interactions are inherently stateful, distinguishing them from traditional stateless web requests (Ghosh, 2025). Agents engage in conversations where each interaction changes the internal state of the agent or the environment. This requires both short-term retention of state (*e.g.*, for each pairwise agent interaction) and long-term state persistence. However, most agentic AI frameworks adopt stateless designs or delegate state management entirely to the developers (Agent Communication Protocol Project, 2025; Google DeepMind, 2025; Marro et al., 2024). Without standardized approaches to state management, agents would remain confined to isolated and context-free exchanges, severely limiting their interoperability and potential to collaborate on complex tasks (Russell & Norvig, 2016).

**Short-term memory through sessions.** Agents require mechanisms to maintain immediate context across multiple message exchanges in a pairwise conversation. This requirement can be satisfied through existing web session management patterns, which are well-established. More specifically, short-term state in web interactions is managed through cookies, which is a standardized approach as part of the HTTP protocol, where a unique session identifier is assigned during the initial interaction and subsequently included in all requests within the same conversation (Peng & Cisna, 2000).

**Long-term memory through databases.** Beyond immediate conversational context, agents often require long-term state storage to store historical interactions and accumulated knowledge across sessions (Sapkota et al., 2025). This long-term memory requirement can be satisfied through established patterns for database integration in web services.

The agents can write to and read from the database as part of their workflow. Similar patterns underpin virtually all data-persistent web applications, from social media platforms to collaborative document editing systems.

Thus, we do not need sophisticated or new solutions to state management in agentic AI. Reusing existing approaches for state management offers advantages over prescriptive standards. It allows agent developers to be sure that all agents will have state management, but leaves them the flexibility of implementing it according to their requirements.

### 5.4. Agent Discovery

Discovery is a foundational requirement for any interoperable agent ecosystem. Without standardized discovery mechanisms, agent ecosystems would fragment into isolated islands of functionality, severely limiting their collective capabilities. Current methods either involve developing isolated networks of agents like Naptha AI (Naptha AI, 2025) or curating a centralized registry of existing AI agents like Agency Marketplace (Agency, 2025). While the former requires all agents to be on the same closed network, the latter requires manual registry self-curation. Both these approaches are not scalable, and hence, unsuitable for WEB OF AGENTS. To enable a truly collaborative WEB OF AGENTS, we need search engines for agents that regularly crawl and index agents on the web. This mirrors how web search engines transformed the early internet from a collection of disconnected pages into a navigable and discoverable information space (Brin & Page, 1998; Schwartz, 1998).

**Unique endpoints.** Agents require globally unique identifiers that can be reliably resolved to network endpoints for other agents or search engines to discover and interact with them. Uniform resource locators (URLs) provide an ideal solution as they are already designed to uniquely identify resources on the web. This approach benefits from the established domain name system (DNS) infrastructure for name resolution without introducing any new identification scheme. This approach provides several advantages: *(i)* URLs are human-readable, *(ii)* they are hierarchical, allowing organizational namespacing of agents, *(iii)* the entire URL resolution infrastructure, including the DNS, has been optimized and maintained to handle the scale of the web, and *(iv)* this design allows agents and webpages to coexist over the web, enabling seamless interoperability.

**Capability advertisement.** Agents must expose metadata about their capabilities and available tools. Most agentic protocols, including A2A, ACP, and agents.json, describe their formal standards for the description of agents and their capabilities. However, the number of fields in such a formal description can range from basic capabilities to guardrails, and is only expected to grow as we build new agentic AI systems (Casper et al., 2025). Therefore, keeping it simple,

each agent should expose a short document that describes its capabilities and tools. This document can potentially be the same as the interaction documentation described in Section 5.2. Similar to A2A, we propose hosting the capability description at a well-known path (similar to *robots.txt* and *sitemap.xml*) (Nottingham, 2019). This standardized location ensures that both search engines and other agents can reliably locate capability information. Importantly, this standardizes the *mechanism* of capability advertisement, not the format. The well-known path convention is itself an extensible IETF standard (RFC 8615) designed to accommodate new use cases over time, meaning future agents may advertise capabilities in different formats without breaking the discovery mechanism.

By adopting these design principles for agent discovery, we establish a minimalistic standardization framework that directly leverages and builds on top of the web's highly optimized and scalable infrastructure. Each component can keep evolving to make WEB OF AGENTS better while maintaining interoperability across agentic AI ecosystems.

### 5.5. Putting it Together

The four building blocks of WEB OF AGENTS form a sufficient foundation for collaborative agentic AI. To illustrate this, consider the AI Scientist MAS (Lu et al., 2024), where specialized AI agents collaborate for scientific discovery. In WEB OF AGENTS, these agents would operate without prior knowledge of other agents and their capabilities. The discovery mechanism regularly indexes all agents. When the hypotheses-generation agent ($H$) needs computational support, it can query the discovery to locate a suitable code-generation agent ($C$) for collaboration. Agent-to-agent messaging enables $H$ to send requests to $C$, while interaction interoperability allows $H$ to understand $C$'s interface requirements and compose properly formatted requests. Finally, state management allows $H$ and $C$ to maintain contextual continuity across multi-turn dialogs without retransmitting complete conversations. Critically, both $H$ and $C$ operate independently and may reside in different agentic ecosystems (*e.g.*, A2A and ACP), yet these four building blocks are minimal and sufficient to enable seamless collaboration. The current nascent state of agentic AI frameworks is a perfect moment for adopting these minimal standards to avoid the *protocol wars* and lay the foundations of an interoperable WEB OF AGENTS.

### 5.6. WEB OF AGENTS in Practice

To demonstrate the feasibility of our proposed WEB OF AGENTS blueprint, we implemented the four building blocks (agent-to-agent messaging, interaction interoperability, state management, and agent discovery) as extensions to existing agentic AI frameworks. Specifically, we forked and

modified the codebases of A2A, ACP, and Agora to incorporate HTTP-based messaging, interaction documentation endpoints, session-based state management, and capability advertisement at well-known paths. Our open-source proof-of-concept implementation required minimal modifications to each framework, typically around 200 lines of code per protocol, demonstrating that adoption of these standards imposes negligible engineering overhead[1]. Furthermore, we ensure that agents within an ecosystem, for example, two agents in the A2A ecosystem, can still operate as intended by the framework developers. The implementation showcases cross-ecosystem agent collaboration where agents implemented in different protocols can discover, interact with, and maintain stateful conversations with one another.

To quantify the impact of WEB OF AGENTS on end-to-end request latency, we run an experiment using one A2A and one ACP agent. With the original code bases of A2A and ACP, the latency overhead, excluding the LLM generation, is $9.17\,\text{ms}$ and $93.04\,\text{ms}$ for A2A and ACP, respectively. With the WEB OF AGENTS code base, these latencies become $10.4\,\text{ms}$ and $96.02\,\text{ms}$, respectively. Since LLM generation takes 3.3 seconds on average in our experiments, the impact of WEB OF AGENTS on latency is negligible.

## 6. Alternative Views

In this section, we will address alternative views that contradict our position.

**Alternative View 1:** *Web technologies were designed for content delivery and should be reinvented for agentic AI.*

We reject the assumption that web technologies are fundamentally incompatible with agentic AI. In fact, web technologies have evolved from serving static websites to dynamic content. At their core, agents are like dynamic websites: they expose some capabilities, respond to structured requests, maintain internal state, and can be discovered and invoked using standard web protocols. Some agentic AI systems are already extending mature web technologies, although in isolation. Thus, agentic AI requires alignment with web technologies, which we have shown is feasible in Section 5.

**Alternative View 2:** *It is too soon to worry about interoperability in Agentic AI: natural selection will eventually determine the best practices and standards.*

While it is true that the agentic AI space is still evolving and diverse designs as well as use cases are expected, deferring interoperability until a mature protocol emerges is shortsighted. History shows that dominant solutions often arise without prioritizing openness (*e.g.*, early cloud APIs or

messaging platforms). Google's A2A may become influential, but it is not inherently neutral nor guaranteed to support open collaboration across providers. Messaging apps serve as a cautionary tale: while dominant platforms emerged, the lack of interoperability led to closed ecosystems, lock-in, and duplicated engineering effort (Yan & Feng, 2023). We thus ask ourselves the question: do we want to monopolize agentic AI by adopting one ecosystem, or allow a decentralized existence of diverse but interoperable ecosystems? Our position takes the decentralized stance: early alignment around minimal standards ensures flexibility and guards against future silos, even if popular solutions emerge later.

**Alternative View 3:** *Standardizing early slows down progress.*

We agree that standardization can slow innovation, especially in a fast-moving field like agentic AI. We acknowledge this and hence, do not advocate for extensive protocols, but for minimal and extensible foundations, just enough to enable interoperability without stifling experimentation. WEB OF AGENTS is the first step towards this. The web itself evolved this way: early standards like HTTP and HTML were minimalistic and were later extended with the uses that were common on the web.

## 7. Final Remarks and Call to Action

In this work, we argue that collaborative agentic AI demands interoperability as a foundational requirement. Rather than waiting for dominant solutions to emerge, we advocate for the adoption of minimal standards that enable agents to discover, communicate, and collaborate across organizational and technical boundaries (see Appendix A). Our proposed WEB OF AGENTS blueprint shows that such interoperability can be achieved by reusing existing web technologies.

Ongoing efforts by AGNTCY, W3C, and The Linux Foundation demonstrate community momentum toward interoperability (see Appendix B for details). To realize an interoperable agentic AI ecosystem, we identify concrete steps for key stakeholders. *Protocol Developers* must align protocols with the four WEB OF AGENTS building blocks, *i.e.*, adopt HTTP as the transport layer, standardize interaction documentation endpoints, implement web session management, and expose capabilities via URLs. *Standards Organizations* like W3C and The Linux Foundation must prioritize these minimal standards as baseline requirements for hosted protocols. Coordinated action is needed now, before fragmentation becomes the norm. By adopting these minimal standards early, we can build the foundations for an open collaborative agentic AI ecosystem.

---

[1] Source code available at `https://github.com/sacs-epfl/web-of-agents`.

## Acknowledgements

This work has been funded by the Swiss National Science Foundation, under the project "FRIDAY: Frugal, Privacy-Aware and Practical Decentralized Learning", SNSF grant number 10001796. We are also grateful to Rasmus Veski for his contributions to the implementation of the WEB OF AGENTS proof-of-concept.

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

## A. Politics and Economics

The case for interoperability in agentic AI extends beyond technical architecture into political and economic domains. Questions of ecosystem fragmentation, vendor lock-in, market concentration, and user autonomy are fundamentally about power distribution, competitive dynamics, and who controls the infrastructure of collaborative AI systems. We acknowledge that interoperability is not merely a technical concern but a sociotechnical one with significant implications for innovation, competition, and access. However, in this position paper, we deliberately take a technical stand. We focus on demonstrating that minimal interoperability standards are architecturally feasible, practically implementable, and sufficient for enabling cross-ecosystem collaboration. This technical foundation is necessary for addressing the broader political and economic considerations. By establishing that interoperability can be achieved through lightweight extensions to existing web technologies, we provide concrete evidence that the technical barriers are surmountable. The political and economic arguments for or against interoperability, while crucial, must be informed by an understanding of what technical solutions are viable and what tradeoffs they entail.

## B. Ongoing Efforts

Our work complements several concurrent efforts toward agentic AI standardization. The W3C AI Agent Protocol Community Working Group is developing specifications for agent communication and interoperability within the web standards ecosystem (W3C AI Agent Protocol Community Group, 2025). The AGNTCY project (Agentcy.org, 2025) provides vendor-agnostic infrastructure for agent discovery, identity, messaging, and observability, enabling cross-framework collaboration. Notably, The Linux Foundation has taken a leadership role in fostering interoperability by hosting both the A2A and ACP protocols under its governance structure, signaling industry commitment to open, collaborative standards development (The Linux Foundation, 2025a;b). Additionally, industry consortia and academic initiatives are exploring security frameworks, trust models, and ethical guidelines for multi-agent systems. Rather than competing with these efforts, our minimal standards approach provides a technical foundation that can accommodate diverse governance models and specialized extensions. We view WEB OF AGENTS as a baseline architecture upon which these complementary initiatives can build, ensuring that efforts toward security, discovery infrastructure, and regulatory compliance operate within an interoperable ecosystem rather than perpetuating fragmentation.

