# OpenReview forum: "Position: Collaborative Agentic AI Needs Interoperability Across Ecosystems"
_ICML.cc/2026/Position_Paper_Track — ICML 2026 Position Paper Track regular_

### Official Review · Reviewer_Luib · 2026-03-10

**Significance:** 3
**Argument Clarity:** 3
**Rating:** 4
**Confidence:** 2

**Questions:**

1. In Section 4, you mentioned that "With interoperability, anyone can publish and run agents that are discoverable and usable by others without needing permission from any authority." Will this cause the issue of intellectual property, since not everyone would like to share their own agents with others?
2. In Section 5.2, you mentioned that "Since modern AI agents are powered by LLMs, they can interpret and adapt to various documentation formats, from structured JSON to natural language descriptions." How do you handle the potential issue of ambiguity in natural language? This is the purpose of introducing structured format in the past.
3. In Section 5.4, you mentioned that "this design allows agents and webpages to coexist over the web, enabling seamless interoperability." Is it possible that agents will incorrectly identify a web page as an agent?

**Alternative Views Section:**

Yes

**Compliance With Llm Reviewing Policy A Conservative:**

Affirmed.

**Discussion Potential:**

3

**Final Justification:**

This position paper is a well-written paper that connects the agentic AI to the matured web technologies. Although I have some concern about the frame it proposes, it does not conceal that this paper can result in some discussion. As a result, it is a good position paper that should be accepted.

**Paper Summary:**

This position paper primarily shows a position that agentic AI should be developed with interoperability by minimal standards borrowed from the matured web technologies. The main contributions of this paper are: (1) it introduces the background of the existing situation of agentic AI, which leads to ecosystem fragmentation that motivates the purpose of this position; (2) it retrospects the history of development from generative AI to collaborative AI to give sufficient background for readers to have the base of the problem; (3) it points out the key issue of AI development: agentic AI solutions are developed in isolation, which may result in incompatibility between ecosystems, and points out the details with concrete examples; (4) To deal with the incompatibility issue, the authors propose to develop interoperability from thw following aspects: security, open participation and scalability; (5) To demonstrate the possibility of interoperability, the authors propose a blueprint to implement it as a web of agents, referring to the ideas from web technologies.

**Position:**

Yes

**Position In Title:**

Yes

**Related Work:**

2

**Strengths And Weaknesses:**

Strengths:
1. The structure of this paper is in general very clear, where the motivation and background knowledge to understand the problem are well written. This can make readers from other fields understand this position paper easily.
2. Most of the claims are equipped with either concrete examples or references, and the logic of clarifying the claims is clear.
3. The blueprint of web of agents is interesting, with the complete plan of how each module should work.
4. The issue that the authors point out is significant enough to incentivise some discussion in the community, since from the perspective of system engineering the development of protocols or how to enable agents to well interact to each other is a cutting-edge problem.
5. The responses to the alternative views are convincing with sufficient evidence from the past experience or good analogies.

Weaknesses:
1. Although I acknowledge that this is a good position paper, I am still confused of if the topic is highly related to machine learning. Agentic AI is no doubt a topic in AI, but it has several aspects. This position paper mainly pinpoints the system engineering level, which I believe is with few connection to machine learning.
2. When talking about collaborative agentic AI, I appreciate that the authors have mentioned the classic multi-agent system research. However, the research trend of multi-agent (reinforcement) learning is missing. This could mislead the readers to exaggerate the position of agentic AI as an emerging breakthrough from the perspective of multi-agent systems.

**Support:**

3

---

> ### Author Rebuttal · Authors · 2026-03-30
>
> We thank Reviewer Luib for the thorough review and the recognition of our paper's clarity and significance.
>
> **W1: Relevance to machine learning.**
>
> We respectfully argue that our work is relevant to the ML and broader ICML community. The individual agents we discuss are powered by LLMs, and collaborative agentic AI is a natural extension of agentic AI. The emerging "protocol wars" in agentic AI will shape how LLM-based agents are built, composed, and deployed. ICML is the right venue for this discussion as it brings together both researchers shaping agentic AI systems and practitioners building and using them.
>
> We acknowledge that our work focuses more on systems for ML. However, systems for ML is explicitly mentioned in the ICML call for papers. Furthermore, the ICML Position Paper Track explicitly invites work that states positions on controversial or important topics, which we believe includes infrastructure and ecosystem design for collaborative agentic AI.
>
> **W2: Missing multi-agent reinforcement learning (MARL) literature.**
>
> We thank the reviewer for this observation. We agree that multi-agent reinforcement learning (MARL) is a rich research area that has contributed foundational ideas to agent coordination, and that RL techniques have been instrumental to the success of modern agentic AI (e.g., RLHF). That said, our paper focuses specifically on the infrastructure and protocol layer for cross-ecosystem interoperability, which is orthogonal to the learning paradigm powering individual agents. We do not intend to position LLM-based agentic AI as a breakthrough over classical multi-agent systems, but rather address a practical systems-level challenge that arises as diverse agents need to collaborate at web scale. We will add a discussion in Section 2, positioning our work relative to MARL (citing, e.g., Zhang et al., "Multi-Agent Reinforcement Learning: A Selective Overview of Theories and Algorithms," 2021).
>
> **Q1: Intellectual property concerns with open agent discovery.**
>
> Discovery of agents and access to agents are distinct concepts. An agent being discoverable does not mean it can respond to all requests. Agents can implement authentication, authorization, and access control, just as websites are discoverable by search engines, but may require a login for access. Our framework is fully compatible with access restrictions; interoperability enables the possibility of interaction, not the obligation. Agent operators retain full control over who can invoke their agents. Therefore, intellectual property concerns are a valid but orthogonal problem.
>
> **Q2: Ambiguity in natural language interaction documents.**
>
> This is a valid concern, but one that modern LLMs are uniquely positioned to handle. Unlike traditional software systems that require specific schemas and parsers, LLM-powered agents can interpret natural language documentation with high fidelity. Our approach allows a spectrum: agents may use structured formats (JSON Schema, OpenAPI) when precision is critical, but Web of Agents does not mandate a single format as a standard. This flexibility is key to adoption: it avoids the premature standardization trap. In practice, we expect high-stakes interactions to use more structured documentation, while simpler interactions may use natural language, similar to how humans communicate. Furthermore, if ambiguity causes a failed interaction, agents can ask clarification questions in natural language (a capability of LLM-powered agents), which is impossible under rigid schema-only approaches. We will discuss this more explicitly in the revision.
>
> **Q3: Can agents incorrectly identify a webpage as an agent?**
>
> This is unlikely in practice. Agents are distinguished from webpages by the presence of a capability document at a well-known path (e.g., `/.well-known/agent.json`), similar to how `robots.txt` distinguishes crawlable sites. A search engine for agents would only index endpoints that expose this specific capability advertisement. Regular webpages without such documents would not be identified as agents. This is a simple, reliable, and already proven pattern on the web.

---

> > ### Author Rebuttal · Reviewer_Luib · 2026-04-01
> >
> > Thanks for the authors' response.
> >
> > About the scope of ML community, I understand your viewpoint. The key point I stand by is that I do not think most agentic AI or LLMs are relevant to ML enough from the traditional perspective of ML, but it does not mean you wrong. As a result, I cannot justify this point at the moment.
> >
> > **"This is unlikely in practice. Agents are distinguished from webpages by the presence of a capability document at a well-known path (e.g., /.well-known/agent.json), similar to how robots.txt distinguishes crawlable sites. A search engine for agents would only index endpoints that expose this specific capability advertisement. Regular webpages without such documents would not be identified as agents. This is a simple, reliable, and already proven pattern on the web."**
> >
> > I can buy this point from the perspective of current application. However, from the general perspective of a multi-agent system this is too narrow, because it only includes a specific implementation at this moment which makes it lose potential to influence the future development (e.g., in the future using files to desribe agents).

---

### Official Review · Reviewer_61L4 · 2026-03-13

**Significance:** 3
**Argument Clarity:** 3
**Rating:** 5
**Confidence:** 4

**Questions:**

1. Can you comment on the weakness listed above?

**Alternative Views Section:**

Yes

**Compliance With Llm Reviewing Policy A Conservative:**

Affirmed.

**Discussion Potential:**

4

**Final Justification:**

The rebuttal did not change my evaluation of Accept.

**Paper Summary:**

The authors argue that current LLM agent technologies risk to evolve into a landscape with incompatible ecosystems, with minimal interaction between the ecosystems. To avoid this the authors propose interoperability techniques that build on ideas developed for web technologies, primarily focusing on agent-to-agent messaging, interaction interoperability, state management and discovery techniques. The authors state that they propose a minimalistic, stripped-down versions of the relatively complex protocols proposed for web services.

**Position:**

Yes

**Position In Title:**

Yes

**Related Work:**

3

**Strengths And Weaknesses:**

Strengths:

- The authors state a clear position, on an issue that is likely going to be contentious in the immediate future. While I personally disagree with this position, I consider it a strenght of the paper, as it can trigger fruitful discussion.
- The authors bring good arguments for the defense of the stated position.
- Alternative views are clearly stated.

Weaknesses:

- Arguably, in a layered networking architecture, the agentic systems sit on top of the application layer. What the paper is proposing is essentially building presentation and session layers on top of this - the implications of these unusual layering are not discussed in the paper.

**Support:**

3

---

> ### Author Rebuttal · Authors · 2026-03-30
>
> We thank Reviewer 61L4 for the positive assessment and the insightful question about network layering.
>
> **W: Unusual layering: agentic systems sit atop the application layer, yet Web of Agents builds presentation/session layers on top.**
>
> We agree that agentic AI systems live at the application layer. We also note that some of the interoperability functions we discuss, most notably interoperability documentation exchange and state management, overlap with concerns historically associated with presentation and session handling. However, our proposal does not introduce these as distinct new layers on top of agentic systems. Rather, they are application-level interoperability functions that collaborative agents must support while reusing existing web infrastructure. We will explicitly clarify this in the paper.

---

> > ### Author Rebuttal · Reviewer_61L4 · 2026-04-03
> >
> > The concern was resolved. I will keep the rating at accept.

---

### Official Review · Reviewer_kZXP · 2026-03-13

**Significance:** 2
**Argument Clarity:** 3
**Rating:** 4
**Confidence:** 3

**Questions:**

While this is a position paper, the proof-of-concept is entirely relegated to the appendix (I understand the page limitation). Modifying A2A and ACP with 200 lines of code shows some feasibility, but lacks rigorous evaluation. Is there no baseline comparison evaluating latency overhead, payload efficiency, or reliability when replacing asynchronous, persistent connections?

**Alternative Views Section:**

Yes

**Compliance With Llm Reviewing Policy A Conservative:**

Affirmed.

**Discussion Potential:**

3

**Final Justification:**

I appreciate the authors' response and their plan to include initial latency and overhead benchmarks in the paper. I decided to hold my positive rating for this paper

**Paper Summary:**

The paper identifies the growing fragmentation among collaborative agentic AI frameworks (e.g., A2A, MCP, ACP) and argues that achieving cross-ecosystem interoperability requires adopting minimal, existing web standards rather than building complex new protocols or brittle translation layers. The authors propose "WEB OF AGENTS," a minimalistic architectural blueprint utilizing HTTP for inter-agent messaging, unconstrained interaction documents for capability sharing, web sessions for short-term memory, and URLs with well-known paths for agent discovery. The core insight is that leveraging web infra prevents ecosystem lock-in and allows better scaling, as demonstrated by a lightweight proof-of-concept integrating multiple disparate frameworks.

**Position:**

Yes

**Position In Title:**

Yes

**Related Work:**

3

**Strengths And Weaknesses:**

## Strength

1. Timely posiiton: The paper captures the current 2024-2026 trend of agentic frameworks (fast growing but frameworks are fragmented, chaos). Addressing the isolation between protocols like A2A, MCP, and ACP is a pressing issue for the deployment of multi-agent systems at scale.
2. Pragmatic and Actionable side: The proposed "WEB OF AGENTS" adheres strictly to the simplicity principle. By advocating for existing, highly scalable web techs rather than enforcing heavy, proprietary orchestration layers, the solution offers immediate applicability.
3. Authors also clearly states the historical failures of monolithic unified protocols (like OSI vs. TCP/IP) and the brittleness of translation layers, further framing its minimalist stance effectively

## Weakness

1. From a structural standpoint, the paper essentially rebrands standard RESTful web architecture as a "novel" agentic foundation. Utilizing HTTP requests for message transport, sessions for state, and URLs for discovery  represents standard software engineering best practices rather than a new architectural paradigm specific to AI.
2. The paper assumes that standard web protocols are entirely sufficient for collaborative agents. However, IMO, it glosses over the fundamental differences between deterministic web services and stochastic LLM behaviors. For instance, handling hallucinated tool calls, streaming intermediate CoT / long cot reasoning states during long multi-agent negotiations, or defining fallback behaviors for ambiguous payloads are critical challenges that standard HTTP requests do not natively solve without additional schema enforcement (which the paper actively discourages).

**Support:**

2

---

> ### Author Rebuttal · Authors · 2026-03-30
>
> We thank Reviewer kZXP for the thoughtful and technically detailed review.
>
> **W1: From a structural standpoint, the paper essentially rebrands standard RESTful web architecture as a "novel" agentic foundation.**
>
> We respectfully clarify that our contribution is not the individual technologies but the position that these existing technologies are sufficient and should be adopted as minimal standards for agentic AI interoperability. As shown in Table 1, none of the analyzed solutions support interoperability with other solutions. Current solutions (A2A, ANP, AITP, LMOS) are building custom protocols, identity layers, and communication mechanisms despite the existence of suitable web technologies. The novelty of our paper lies in:
> - *The systematic diagnosis* of ecosystem fragmentation across eight prominent frameworks.
> - *Arguing for minimal standards* as a viable solution to ecosystem fragmentation.
> - *The sufficiency argument* that exactly four building blocks, all achievable via existing web technologies, are necessary and sufficient for interoperability.
> - *The proof-of-concept*, demonstrating this across three real-world frameworks (A2A, ACP, Agora) with minimal code changes.
>
> The fact that the solution is "software engineering best practices" strengthens our position: it means the barrier to interoperability is not technical but organizational, making the call to action all the more urgent. This is precisely the point of a position paper.
>
> **W2: Glossing over differences between deterministic web services and stochastic LLM behaviors.**
>
> This is a perceptive observation. However, we respectfully argue that these challenges are orthogonal to the interoperability layer:
>
> *Hallucinated tool calls*: This is an internal reliability problem in agents, not an inter-agent protocol problem. The correct mitigation lives at the agent level (validation, guardrails, retry logic), not the transport level. The interoperability layer neither prevents nor introduces hallucinations.
>
> *Streaming intermediate CoT/reasoning*: HTTP, the protocol underpinning the modern web, already supports streaming via chunked transfer encoding and server-sent events, both mature and widely deployed. A2A itself uses SSE for streaming, which runs over HTTP, reinforcing our argument that HTTP is sufficient. We deliberately avoid mandating streaming as part of the minimal standard because not all agents require it, but nothing in our architecture prevents its use.
>
> *Ambiguous payloads*: Our interaction document mechanism (Section 5.2) directly addresses this by allowing agents to document expected inputs, formats, and constraints. This is strictly more flexible than rigid schema enforcement, because LLM-powered agents can interpret documentation in natural language or structured formats. Schema validation can be layered on top as needed. Our minimal standard does not preclude it but avoids mandating a single schema standard on agentic communication frameworks prematurely.
>
> We will add a discussion of how stochastic LLM behaviors interact with our proposed building blocks in the revision.
>
> **Q: Lack of rigorous evaluation (latency, payload efficiency, reliability).**
>
> We acknowledge this limitation. However, we note:
>
> - *This is a position paper*, and the ICML Position Paper Track explicitly values argumentation and discussion potential over empirical evaluation.
> - *The overhead is architecturally predictable*. Replacing JSON-RPC (used by A2A) with plain HTTP REST adds no additional round-trips; both operate over TCP. The payload difference is negligible (a few bytes of header). The interaction document is fetched once per agent pair. Most importantly, the protocol overhead is dwarfed by LLM inference latency (typically 1–30s per agent turn vs. <10ms protocol overhead).
> - *Our proof-of-concept demonstrates functional equivalence*: agents across A2A, ACP, and Agora successfully discover, communicate, and maintain state with each other. The ~200 lines of modification per framework further demonstrates minimal engineering cost.
>
> We plan to include initial latency and overhead benchmarks in the camera-ready appendix. Furthermore, to further demonstrate functional equivalence, we plan to open-source a demo of agents in the Web of Agents collaborating across ecosystems.

---

> > ### Author Rebuttal · Reviewer_kZXP · 2026-04-04
> >
> > Thanks for the response. I hold my positive rating for this paper.

---

### Official Review · Reviewer_LnaB · 2026-03-13

**Significance:** 3
**Argument Clarity:** 3
**Rating:** 5
**Confidence:** 3

**Questions:**

1. See Weakness 1.

2. What are the possible issues if a web-based protocol is used? Will that raise concerns about privacy or communication efficiency?

**Alternative Views Section:**

Yes

**Compliance With Llm Reviewing Policy A Conservative:**

Affirmed.

**Discussion Potential:**

2

**Final Justification:**

My primary concern is addressed in the rebuttal as the authors provide a clear explanation of the necessity of the web-based protocols as opposed to existing communication protocols.

**Paper Summary:**

Current AIs are heterogeneous. To allow collaboration at a large scale, this paper proposes a web-based protocol that enables communication, interaction, state/memory storage, and the discovery of other agents. The authors also argue that a minimal protocol should be favored.

**Position:**

Yes

**Position In Title:**

Yes

**Related Work:**

3

**Strengths And Weaknesses:**

Strengths:

1. This paper is well motivated: a unified, lightweight communication protocol between different LLM agents enables many benefits.

2. The ecosystem of collaborative LLMs is of high significance.

Weaknesses:

1. This paper does not answer the question of why letting AIs use communication software or platforms cannot be an obvious answer to the problem of incompatibility. What is the reason that this obvious solution does not work?

2. The proposed methods to build a Web of Agents are relatively primitive. However, as the main focus of the paper is to position such protocols, this weakness is minor.

**Support:**

3

---

> ### Author Rebuttal · Authors · 2026-03-30
>
> We thank Reviewer LnaB for recognizing the motivation and significance of our work. We address the weaknesses and questions below.
>
> **W1: Why can't existing communication software/platforms solve incompatibility?**
>
> This is a valid question. We argue that existing communication platforms are insufficient as a foundation for Web of Agents for the following three reasons.
>
> Firstly, our paper argues for having minimal standards. A platform implements these standards. For example, SMTP is a standard, and Gmail is a platform implementing the SMTP standard. The power of these ecosystems comes from the fact that any platform implementing the standard can interoperate with any other. In contrast, Slack, Discord, or Microsoft Teams are proprietary platforms with proprietary standards. Agent builders who adopt one are locked into that provider's ecosystem. This is precisely the lock-in dynamic we warn against in Section 4.
>
> Secondly, using existing communication platforms doesn’t solve the problem of agent interoperability on its own. As we point out in Section 5 (and visualize in Figure 3), we also require interaction interoperability, state management, and agent discovery. Adapting existing platforms for this purpose would require either (1) accepting their constraints and limitations, or (2) building so much additional infrastructure on top of them that the platform merely becomes a transportation layer. At that point, it is more viable to standardize the underlying layer directly.
>
> Thirdly, existing communication software can change APIs, impose rate limits, or even discontinue services. For instance, WhatsApp's API rate limits and policy changes have historically disrupted bots and integrations built on top of it. This makes them a fragile foundation for multi-agent infrastructure.
>
> **W2: The proposed methods to build a Web of Agents are relatively primitive.**
>
> We appreciate the reviewer's acknowledgement that this is minor, given the paper's nature. We intentionally advocate for minimal standards (KISS principle, see Section 5), as history shows that overly complex early standards (e.g., OSI model) fail to gain adoption. The primitiveness is a feature, not a bug: it lowers the adoption barrier while remaining extensible. The web itself succeeded precisely because HTTP (and HTML) were initially "primitive."
>
> **Q2: What are the possible issues if a web-based protocol is used? Will that raise concerns about privacy or communication efficiency?**
>
> Web-based protocols inherit the same challenges as any web service, primarily around privacy and efficiency. However, we argue these are largely addressed by decades of web infrastructure maturation.
>
> *Privacy*: HTTP supports end-to-end encryption via TLS, OAuth-based authentication, and fine-grained access control. These are all mature, battle-tested mechanisms that are more scrutinized than the custom security mechanisms some agentic frameworks introduce. We discuss the security benefits of standardization in Section 4. We acknowledge that agent-specific privacy concerns (e.g., preventing unintended data leakage across agent interactions) deserve further study, and we will expand this discussion in the revision.
>
> *Efficiency*: Modern HTTP versions have addressed historical efficiency concerns. Moreover, for the vast majority of agentic interactions (which involve natural language payloads orders of magnitude larger than protocol overhead), HTTP introduces negligible overhead compared to more specialized protocols like JSON-RPC. The dominant cost in agent interactions is LLM inference (typically 1–30 seconds per agent turn), not transport (<10ms).

---

> > ### Author Rebuttal · Reviewer_LnaB · 2026-04-06
> >
> > My concerns are fully solved and I would increase my recommendation to 5.

---

### Decision · Program_Chairs · 2026-04-30

**Decision:**

Accept (regular)

**Comment:**

The reviewers found the paper to be timely and agreed with its central premise.  No major flaws surfaced in the review phase. The most negative comments were about whether the paper is really an AI paper as opposed to a systems engineering paper (I certainly think it is an AI paper, it's about AI agents afterall) and a comment about novelty. But here they don't mean novelty in terms of the position itself, which surely is novel, they mean novelty with respect to the fact that all the technologies recommended in the position are standard things that have been around for decades and deeply embedded in the architecture of the internet. Like the authors and their "keep it simple stupid" slogan, I agree that this is the right approach. I see the paper as the authors taking the opportunity in the present moment in AI history to remind our community of things that the internet architecture systems engineering community discovered long ago. This is good though since these ideas are not commonly known in AI, and the paper does a very good job translating their context to the new problem at hand: AI agents.